# Autozygosity influences cardiometabolic disease-associated traits in the AWI-Gen sub-Saharan African study

Francisco C. Ceballos 1, Scott Hazelhurst 1,2, David W. Clark3, Godfred Agongo1,4, Gershim Asiki 5, Palwende R. Boua 1,6,7, F. Xavier Gómez-Olivé8, Felistas Mashinya9, Shane Norris1,6, James F. Wilson 3,10 & Michèle Ramsay 1,6✉

The analysis of the effects of autozygosity, measured as the change of the mean value of a trait among offspring of genetic relatives, reveals the existence of directional dominance or overdominance. In this study we detect evidence of the effect of autozygosity in 4 out of 13 cardiometabolic disease-associated traits using data from more than 10,000 sub-Saharan African individuals recruited from Ghana, Burkina Faso, Kenya and South Africa. The effect of autozygosity on these phenotypes is found to be sex-related, with inbreeding having a significant decreasing effect in men but a significant increasing effect in women for several traits (body mass index, subcutaneous adipose tissue, low-density lipoproteins and total cholesterol levels). Overall, the effect of inbreeding depression is more intense in men. Differential effects of inbreeding depression are also observed between study sites with different night-light intensity used as proxy for urban development. These results suggest a directional dominant genetic component mediated by environmental interactions and sex-specific differences in genetic architecture for these traits in the Africa Wits-INDEPTH partnership for Genomic Studies (AWI-Gen) cohort.

[1] Sydney Brenner Institute for Molecular Bioscience, Faculty of Health Sciences, University of the Witwatersrand, Johannesburg, South Africa. [2] School of Electrical & Information Engineering, University of the Witwatersrand, Johannesburg, South Africa. [3] Centre for Global Health Research, Usher Institute, University of Edinburgh, Teviot Place Edinburgh EH8 9AG, UK. [4] Navrongo Health Research Centre, Navrongo, Ghana. [5] African Population and Health Research Center, Nairobi, Kenya. [6] Faculty of Health Sciences University of the Witwatersrand, Division of Human Genetics, National Health Laboratory Service and School of Pathology, Johannesburg, South Africa. [7] Clinical Research Unit of Nanoro, Institut de Recherche en Sciences de la Santé, Nanoro, Burkina Faso. [8] MRC/Wits Rural Public Health and Health Transitions Research Unit (Agincourt), School of Public Health, Faculty of Health Sciences, University of the Witwatersrand, Johannesburg, South Africa. [9] Department of Pathology and Medical Science, School of Health Care Sciences, Faculty of Health Sciences, University of Limpopo, Polokwane, South Africa. [10] Medical Research Council Human Genetics Unit, Institute of Genetics and Molecular Medicine, University of Edinburgh, Western General Hospital, Edinburgh EH4 2XU, UK. ✉email: Michele.Ramsay@wits.ac.za

Obesity and its associated cardiometabolic diseases (CMDs) have been rapidly increasing in sub-Saharan Africa. The continent is in the turmoil of an epidemiological transition characterized by complex patterns of change in health and diseases interacting with demographic, dietary, economic and social determinants[1]. However, this health and demographic transition is very heterogeneous across the African continent and varies dependant on the epidemiological transition at the site. The main purpose of the AWI-Gen (Africa Wits-INDEPTH partnership for Genomic studies) H3Africa Consortium study is to examine the prevalence of CMD-associated risk factors, regional burden, and to explore gene-gene and gene-environment interactions that contribute to disease risk[1,2]. To achieve its goals, AWI-Gen collected genotype and phenotype data on more than 10,000 individuals from rural and urban sites across four sub-Saharan African (SSA) countries (Fig. 1). Biomarkers of CMD included six anthropometric traits (height, weight, body mass index (BMI), waist-to-hip ratio (WHR) and visceral and subcutaneous adipose tissue (VAT and SCAT)), four classical lipid traits (total-cholesterol (TC), high density lipoprotein cholesterol (HDL), low density lipoprotein cholesterol (LDL) and triglycerides (TG)) and three circulatory traits (pulse rate, diastolic and systolic blood pressure). The objective of this study is to learn more about the genetic architecture of CMD-associated traits in SSA by analyzing the contribution of inbreeding depression (ID) to these traits.

Dissecting the genetic architecture of complex traits provides a deeper understanding of disease etiology and insights that could contribute to screening, diagnosis, prognosis and therapy[3]. ID, manifest by an effect of genomic homozygosity on phenotypic values, implies some degree of directional dominance or overdominance in the genetic architecture[4]. ID is influenced by two main factors: the amount of inbreeding and thus auto-zygosity; and the degree to which the dominance of causal loci is biased in one direction (which can be mainly caused by selection pressure[5–8]). For this study, the inbreeding coefficient is calculated from genomic data through the analysis of *runs of homozygosity* (ROH: genomic tracts where homozygous markers occur in an uninterrupted sequence)[9]. African genetic homozygosity is particularly understudied, e.g., 30,000 individuals of African ancestry were included within the largest (~1.4 M) published study, and of those, only 1000 were resident in SSA itself, all of whom were urban dwellers[10]. Eleven traits in total are shared by this analysis and the one performed by Clark et al. 2019, with our study replicating results for height, weight, total cholesterol, LDL, BMI, and triglycerides levels and presenting a statistically-powered ROH analysis of Africans.

## Results and discussion

**Inbreeding in the AWI-Gen cohort.** The mean genomic inbreeding coefficient, $F_{ROH}$, for the entire AWI-Gen cohort is 0.0093, equivalent to their parents being on average about second cousins, once removed, but likely generated in many cases from the sum of multiple more distant relationships. No differences in the $F_{ROH}$ were found between men and women or between urban and rural sites ($F_{ROH}$ men = 0.0045, women = 0.0049) ($F_{ROH}$ urban sites = 0.0051, rural sites = 0.0049).

**Traits affected by Inbreeding Depression.** The substantial levels of genomic homozygosity exhibited by the AWI-Gen cohort allow us to explore their effect on 13 CMD-associated traits and their relationship to socio-economic status (Fig. 2a). We detected

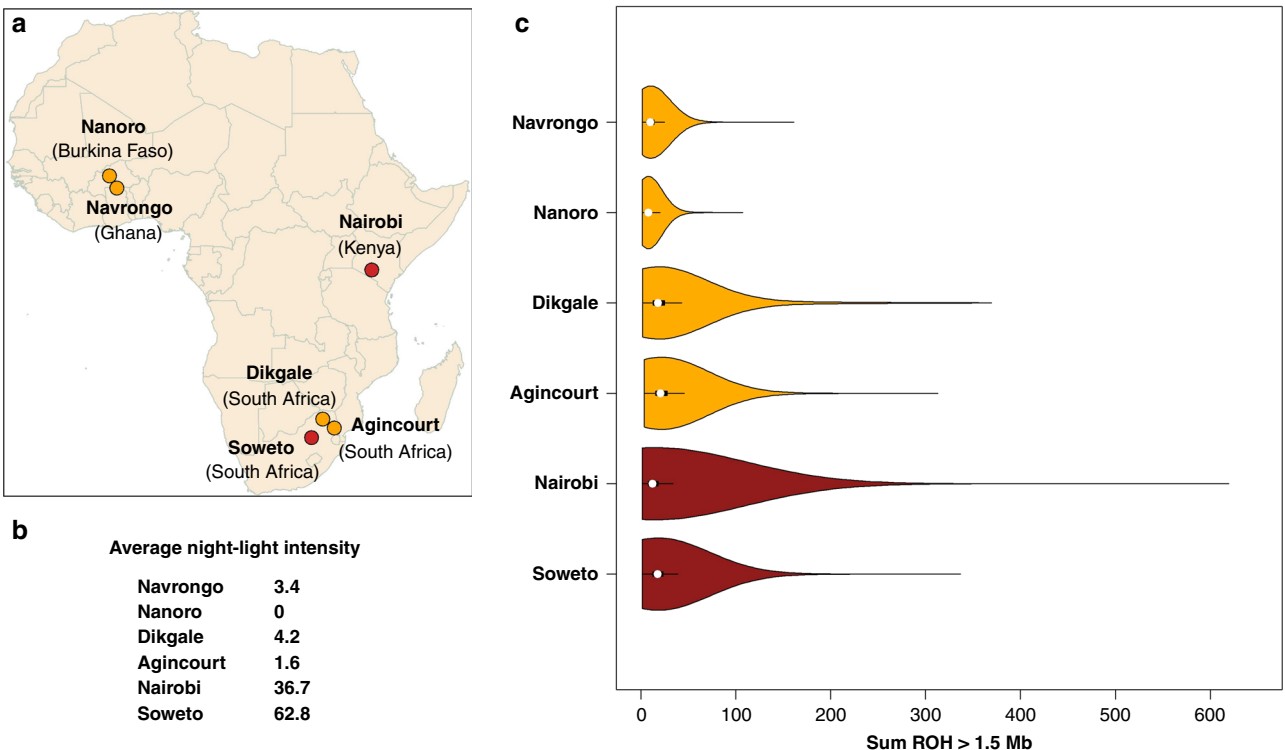

**Fig. 1 Recruiting AWI-Gen study sites, night-light intensity and distribution of the sum of ROH (ROH > 1.5 Mb). a** Locations of the recruitment sites for the AWI-Gen study. Nanoro (Burkina Faso) N = 1,983; Navrongo (Ghana) N = 1,844; Nairobi (Kenya) N = 1,771; Dikgale (South Africa (S.A.)) N = 1,143; Agincourt (S.A.) N = 2,094; Soweto (S.A.) N = 1,782. **b** Economic and urban development is proxied by 2013 night-light intensity data made available by the National Geophysical Data Centre (NGDC) of the National Oceanic and Atmospheric Administration of the U.S. Night-light values range from 0 (unlit) to 63. Sites with average light-night intensity below and above 5 are colored orange and red respectively. **c** Violin plot showing the distribution of ROH longer than 1.5 Mb for each study site. Medians with the interquartile range (IQR) are shown in Supplementary Table 7.

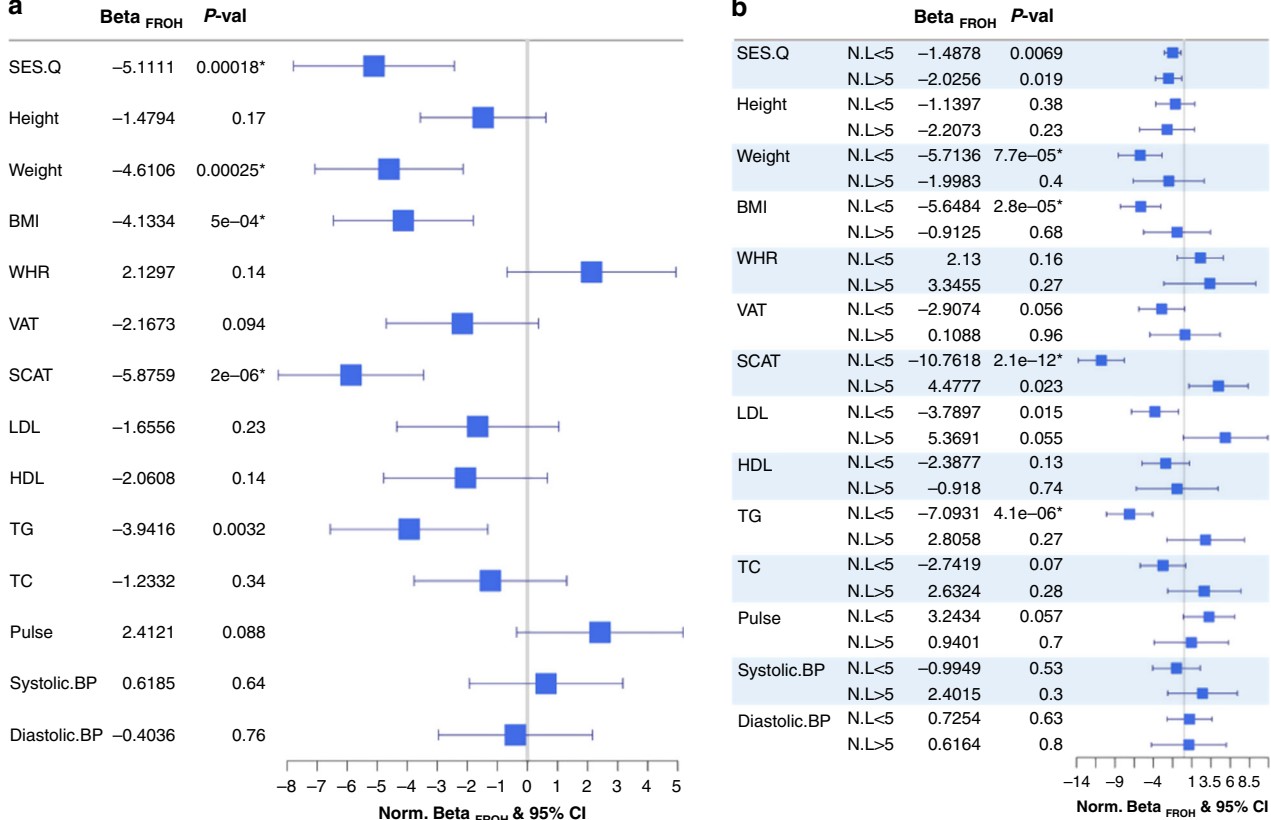

**Fig. 2 Effect of inbreeding depression on cardiometabolic disease-associated traits in the AWI-Gen cohort.** To allow comparison between traits, effects estimates ($\beta_{FROH}$) and 95% confidence intervals are shown in units of within-group standard deviations. A regular Bonferroni correction for multiple testing was applied; significance is shown with an asterisk (0.05/14). **a** Normalized ($\beta_{FROH}$) and 95% confidence intervals on 14 quantitative traits known to be cardiometabolic disease-associated traits: socio-economic status (SES.Q); six anthropometric traits, height, weight, body mass index (BMI), waist-to-hip ratio (WHR), visceral adipose tissue (VAT) and subcutaneous adipose tissue (SCAT); four classical blood lipids, high and low density lipoprotein cholesterol (HDL, LDL), total cholesterol (TC), and triglycerides (TG); and three circulatory traits, pulse rate, systolic blood pressure (Systolic BP) and diastolic blood pressure (Diastolic BP). Sample sizes, effect estimates in measurement units, standard deviations and p-values are shown in Table S1. **b** Differences in the effects of normalised ($\beta_{FROH}$) and 95% confidence intervals between sites with an average night-light intensity below and above 5. Sample sizes, effect estimates in measurement units, standard deviations and p-values for dwellers living in sites with average night-light intensity below and above 5 are shown in Tables S2 and S3 respectively.

a negative association of inbreeding depression with socio-economic-status within each site; in general, having a higher $F_{ROH}$ was associated with a lower socio-economic-status. When the entire cohort is considered, we detected significant negative ID for weight, BMI and SCAT. However, the intensity of the ID for these traits was not very strong and the offspring of the equivalent of a second cousin mating had on average, a reduction of 0.51 kg, 0.18 kg/m², and 0.049 cm, respectively. In a previous study using ~1.4 M individuals worldwide, the average reduction in weight and BMI by ID was estimated as 0.85 kg and 0.25 kg/m², respectively[10]. As a replication comparison, we can conclude that the effect of the inbreeding depression has the same direction for the 11 traits shared by Clark et al.[10] and this study. Furthermore, among the African-descendant individuals analyzed by Clark et al.[10] (Supplementary Data 18) inbreeding depression for height, weight, waist to hip ratio and BMI was found by both studies (Supplementary Table 1 and 2 in this study).

**Urbanization-specific effects of ID.** In SSA, stratifying by night light intensity (luminosity) as a proxy for urban development (see Methods) revealed the effect of ID on the traits is more intense in less developed more rural study sites (Fig. 2b). The average reduction in trait value for offspring of the equivalent of

a second cousin mating was greater for each of the significantly associated traits in areas with less luminosity (0.70 kg in weight, 0.26 kg/m² in BMI and 0.081 cm in SCAT). Moreover, for TG levels that were not significant after multiple test correction in the overall analysis, there is a strongly significant ID effect in the study sites with less luminosity (0.024 mmol/L). As the sample size for the less developed sites is almost double that for more urban sites, we have more power to detect the effects of ID there. However, the effect sizes for ID on SCAT, LDL and TG in more urban sites were significantly different from less developed sites ($P = 2.2 \times 10^{-16}$, $2.2 \times 10^{-5}$ and $3.2 \times 10^{-9}$ respectively, using standard ANCOVA), interestingly showing that the effects in less developed sites were in the opposite direction for these traits (Fig. 2b). We suggest that environmental changes affecting lifestyle factors may be playing a role since food resources, comfort commodities and even primary healthcare are limiting factors in less developed sites. The results indicate that the differences in the effects of ID between more and less developed study sites are not due to unmeasured confounding variables, since we observed a negative effect of autozygosity on socio-economic-status (SES) in both groups (of similar strength), that is, increasing autozygosity was associated with decreased SES. Further, we accounted for the effect of SES by including it as a covariate in all ID analyses (see Methods). In order to ameliorate potential

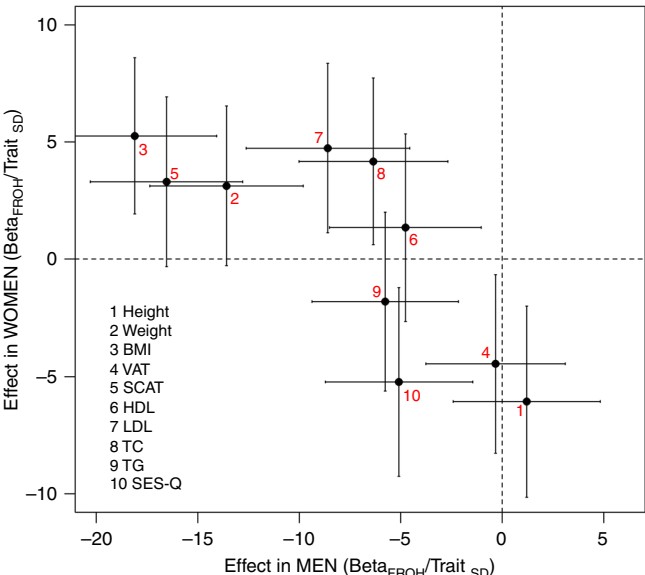

**Fig. 3 Sex-specific inbreeding depression.** As in the previous figure effects estimates and coefficients were standardized using within-sex standard deviations. Effects estimates ($\beta_{FROH}$) and 95% confidence intervals are shown in units of within-sex standard deviations. The effect of $F_{ROH}$ in men versus that in women is shown for 9 traits for which there was evidence of significant differences in the effects between sexes. Traits such as body mass index (BMI), subcutaneous adipose tissue (SCAT), low density lipoprotein cholesterol (LDL) and total cholesterol (TC) showed sex-specific effects of opposite direction, positive in women but negative in men. For just two traits, visceral adipose tissue (VAT) and height, the magnitude of effect was greater in women.

cultural bias in SES measurement and other differences among populations, we used the quintiles of the SES variable (SES.Q) calculated within each site independently (see Methods section Trait Definition).

**Sex-specific effects of ID**. When the sexes were considered separately, we observed a sex-specific effect of inbreeding. For BMI, SCAT, LDL and TC, inbreeding had a decreasing effect in men but an increasing effect in women (Fig. 3). Overall, ID was significantly more intense in men, after normalizing the ID effect sizes for each trait (see Methods). The mean effect across all traits of a homozygous genome ($\beta_{FROH}$) was $-4.9 \pm 5.9$ Trait$_{SD}$ in men and only $1.1 \pm 3.8$ Trait$_{SD}$ in women (Mann-Whitney U test $p = 0.0155$). Men born from a second cousin marriage would be, on average, 2.26 kg lighter, have a reduction of 1.01 kg/m$^2$ in BMI and 0.18 cm less SCAT. However, women would be, on average, 2.1 kg heavier, have an increase of 0.84 kg/m$^2$ in BMI and 0.15 cm more SCAT (however, $\beta_{ROH}$ in women were non-significant, Fig. 3). Also, women born from second cousin marriage would be 5.2 mm shorter, somewhat shorter than estimates found by other studies (average reductions of 2.9 mm[10], 3.0 mm[11] and 1.7 mm[12] for second cousin offspring). Sex-specific ID effects for all the traits shared with Clark et al.[10] (weight, BMI, total cholesterol, triglycerides and LDL) had the same directions, thus replicating their outcomes. Our results further suggest different autosomal genetic architectures, interactions or unmeasured biases for the CMD-associated traits considered in this study for men and women.

**ID is caused by rare recessive variants**. We further explored whether inbreeding depression was caused by common or rare variants by comparing the effect of the $F_{ROH}$, genomic inbreeding coefficient using the genomic relationship matrix ($F_{GRM}$) (see methods section) and $F_{outsideROH}$. We first fitted bivariate models with $F_{ROH}$ and $F_{GRM}$ as explanatory variables and then we explored the effect of $F_{outsideROH}$, which is not an estimator of the inbreeding coefficient, but describes the homozygosity of common SNPs outside ROH. As can be seen in Supplementary Table 4, for all the traits that were significant in the overall univariate analysis (SES.Q, weight, BMI, SCAT and TG) we find that $\hat{\beta}_{F_{ROH}|F_{GRM}}$ is of a greater magnitude than $\hat{\beta}_{F_{GRM}|F_{ROH}}$ in the conditional analysis. Furthermore, for all of the traits but SCAT, $\hat{\beta}_{F_{GRM}|F_{ROH}}$ does not differ from zero, thus indicating that for these traits the variation of $F_{GRM}$ is not associated with any change in trait values. Like $F_{GRM}$, $F_{outsideROH}$ which captures common SNPs in strong linkage disequilibrium (LD), was not significant for any trait or any model (Supplementary Tables 1–3). These results suggest that autozygous rare recessive variants found in ROH, rather than homozygous common variants in strong LD, are causing the inbreeding depression and this is consistent with the dominance hypothesis[4,13]. The fact that the bivariate models show that $F_{ROH}$ captures the signal better than $F_{GRM}$ and $F_{outsideROH}$ also suggests that in this example inbreeding depression is caused by directional dominance. Overdominance—the positive selection on heterozygotes bringing alleles to intermediate frequency—would predict that more common homozygous SNPs outside ROH would also have an effect.

Two recent studies found evidence for the same conclusion. Clark et al.[10], analyzing a large sample size of 1.4 M found that for all the traits affected by inbreeding depression $\hat{\beta}_{F_{ROH}|F_{GRM}}$ was of greater magnitude and in many cases $\hat{\beta}_{F_{GRM}|F_{ROH}}$ was also non-significant. Our results therefore confirm what Clark et al.[10] and Johnson et al.[14] found.

**Genomic regional $F_{ROH}$ effects**. We tested whether the genome-wide ID effects we observed came from a small number of major loci with large effects or resulted from the polygenic effect of many loci with small effects. We thus divided the genome into 1,000 non-overlapping 3 Mb-wide windows and assessed whether homozygosity of these regions was associated with the CMD-associated traits (see Methods). Only two traits were suggestively significantly influenced by individual regions: BMI in men (Fig. 4a, Supplementary Table 5) and VAT in women (Fig. 4b, Supplementary Table 6). For all other traits we have no evidence of major loci that may be exerting large effects, rather the ID appears to be polygenic in origin. For BMI in men, a significant window on chromosome 14 was detected, including 23 protein-coding genes (Supplementary Table 5). Published genome-wide association studies (GWAS) have reported associations with BMI in two of these genes (KCNH5[15] and FUT8[16]), which are known to be pleiotropic. KCNH5 (potassium voltage-gated channel subfamily H member 5) has been found to be associated not just with BMI but also with SCAT[17], intelligence[18], and sleep duration[19]. FUT8 (fucosyltransferase 8) has also been associated with IgG glycosylation patterns[20], age at menarche[21], schizophrenia[22], plasma N-glycans[23], head circumference[24], the plasma proteome[25], multiple sclerosis[26] and systolic blood pressure[16]. Thus, the strongest directionally dominant effects on BMI are shared with loci also imparting additive effects discovered by GWAS. Two windows on chromosomes 2 and 4 were found to be associated with VAT in women (Supplementary Table 6). No GWAS hits were previously found for this trait in these regions; however both regions include genes with multiple associations with other traits.

In this study we analysed the effect of inbreeding depression on 13 cardiometabolic disease-associated traits in the AWI-Gen

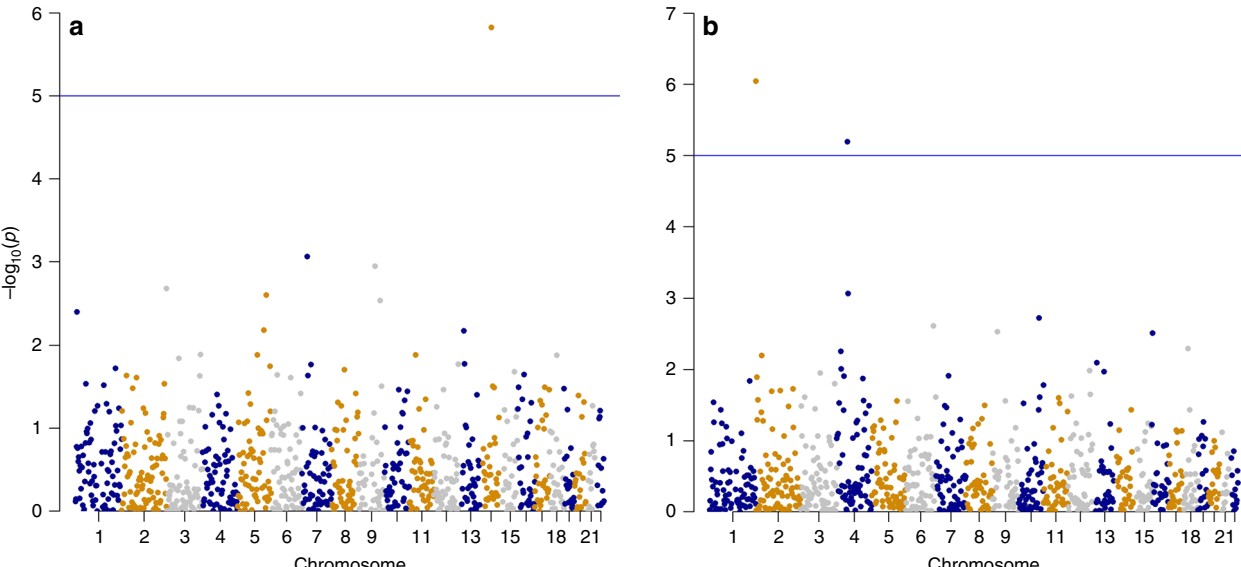

**Fig. 4 Inbreeding depression in genomic windows.** ID was measured in 1000 three Mb-wide non-overlapping sliding windows along the genome; the $p$-value threshold was corrected by Bonferroni and set at $< 1 \times 10^{-05}$. **a** Inbreeding depression on body mass index (BMI) for men. A window on chromosome 14 (position 63-66 Mb) was highly significant. 23 protein coding genes were found in this window as shown in Table S5. Two SNPs located in this window, in *KCNH5* and *FUT8*, were associated with BMI in GWAS studies. **b** Inbreeding depression on visceral adipose tissue (VAT) for women. Two windows on chromosomes 2 (position 0-3 Mb) and 4 (position 57-60 Mb) were significant. 9 protein coding genes were found in the chromosome 2 window and 2 on chromosome 4 (Supplementary Table 4). No GWAS hits related to this trait have been reported in these regions of chromosome 2 and 4.

cohort with the intention of better understanding the genetic architecture of these traits in sub-Saharan Africa. We found significant evidence for inbreeding depression for four phenotypes, which was strongly enhanced in less developed rural regions and also stronger in men. Our results suggest a complex genetic architecture for these traits with interactions by rurality. Larger studies of African populations will further illuminate the factors contributing to complex disease risk in sub-Saharan African populations.

## Methods

**Population genetics overview.** Our objective was to explore the directionally dominant component of the genetic architecture of 13 cardiometabolic disease (CMD) associated traits measured in 10,776 sub-Saharan African (SSA) individuals recruited by the AWI-Gen study. In order to achieve this, we measured the effect of inbreeding depression on these risk factors. In classical population genetics inbreeding depression (ID) is defined as the reduction of the mean fitness in a population because of inbreeding[27]. Currently this definition has been generalized for any complex trait as the change in the mean phenotypic value in a population because of inbreeding. Considering the combined effect of all the loci that affect a character, as far as the genotypic value of the loci combined additively is concerned, the mean character value of a population with inbreeding coefficient ($F$) is given by:

$$M_F = M_0 - 2F \sum d_i \bar{p}_i \bar{q}_i.$$ (1)

where $M_0$ is the population mean before inbreeding and p and q are allele frequencies. The change of the mean on inbreeding is therefore[5]

$$-2F \sum d \bar{p} \bar{q}$$ (2)

This shows that inbreeding will change the mean value of a character in a population when the sum of the genotypic value of the heterozygotes ($d$) is different from 0 (i.e., the character needs to exhibit directionally dominant (or overdominant) genetic architecture[4]). Further, when loci are combined additively, the change of the mean on inbreeding should be directly proportional to the coefficient of inbreeding[4]. This allows us to detect inbreeding depression in complex characteristics exhibiting directional dominance using regression analysis as long as the population under study practices some inbreeding. It is important to realize that the genetic architecture of a characteristic, including the effects of ID, is not necessarily constant among different populations. The intensity of the ID and the genetic architecture of a trait depend on the selection pressure, environmental factors and population structure inasmuch as genetic frequencies change between populations. The considerable burden of long ROH (>1.5 Mb) present in some sub-Saharan populations[28] provides the opportunity to test whether CMD-associated traits gathered by the AWI-Gen initiative exhibit inbreeding depression.

**AWI-Gen Cohort.** The Africa Wits-INDEPTH Partnership for Genomic Studies (AWI-Gen) is an NIH-funded Collaborative Centre of the Human Heredity and Health in Africa (H3Africa) Consortium[2]. It is a partnership between the University of the Witwatersrand and the International Network for Demographic Evaluation of Populations and Their Health (INDEPTH) that includes Health and Demographic Surveillance System (HDSS) Centres and the Developmental Pathways for Health Research Unit (DPHRU), that have longitudinal cohorts in Navrongo, Ghana; Nanoro, Burkina Faso; Nairobi, Kenya; Agincourt, South Africa; Dikgale, South Africa and Soweto, South Africa. In total, over 12,500 participants were recruited between August 2013 and August 2016, of which 10,776 were aged 40–60 years[1]. Pregnant women, first-degree relatives, recent immigrants and individuals with physical impairments preventing the measurement of different characteristics were excluded. The objectives of the AWI-Gen study can be summarized as follows: (1) To build capability for genomic research in these centers and countries by providing opportunities to develop skills including biostatistics, genomics, and bioinformatics. (2) To understand population structure and genetic architecture among recruiting sites in order to inform analysis strategies and evaluate impact across different ethnicities. (3) To understand the prevalence and genetic basis of cardiometabolic diseases (CMD)[2]. The study was approved by the Human Research Ethics Committee (Medical) of the University of the Witwatersrand (Wits) (protocol numbers M121029 and M170880), and each contributing Centre obtained additional local ethics approval as required. Data and samples were collected following community engagement and individual informed consent[1].

Biomarkers of CMD gathered by the AWI-Gen study include anthropometric variables including height, weight, body mass index (BMI), waist-to-hip ratio (WHR) and fat distribution (visceral adipose tissue (VAT) and subcutaneous adipose tissue (SCAT)); lipid composition variables including low density lipoprotein cholesterol (LDL), high density lipoprotein cholesterol (HDL), total cholesterol (TC) and triglycerides (TG); circulatory traits including pulse rate and blood pressure (systolic and diastolic). Besides the above biomarkers a socio-economic-status variable was also calculated for each individual. This variable (SES) was obtained by adding the items a person has in their household from among a list defined items, determined as appropriate for their country. The SES quintiles for each study site were calculated and individuals assigned to a quintile. This was used in the analysis to remove a potential cultural component related to socio-economic-status as is shown in[29]. See Trait Definition section.

AWI-Gen individuals were genotyped using the H3Africa Custom Genotyping Array, which is a high-density genotyping array (2.267 million SNVs) designed by an H3Africa project team to maximize capturing common variation in African population. Manufactured by Illumina, the array has been used by several H3Africa projects. Details of the array and the SNVs captured can be found at https://www.h3abionet.org/h3africa-chip.

A total of 11,076 samples were genotyped. We removed duplicate and non-autosomal SNVs; SNVs with genotype missing rate > 0.01; minor allele frequency < 0.01 and SNVs that deviated from Hardy-Weinberg equilibrium test

($p$-value $< 5.0 \times 10^{-4}$). Similarly, at the sample level, individuals with genotype missing rate $> 0.02$, that failed the sex check and one of the individuals from each related pair (PIHAT $> 0.8$—potential duplicates) were removed. Also, 159 first degree relatives were removed from the Agincourt site. These QC steps were performed using the H3Africa GWAS pipeline[30] and resulted in the final dataset containing 1,733,121 SNVs and 10,617 individuals.

**Trait definition**. 14 different traits and different control variables like age, sex, site of sampling, education and occupation were also considered in the analysis. All traits are defined below under headings in the format (short name –) full name—units. Further details can be found in Ali et al.[1].

Socio-economic-demographic variables

**age—years**. Age at data collection (calculated using the self-declared date of birth and the date of the interview with the individual).

**sex**. Sex was self-reported and validated by genomic data.

**site—Site of sampling**. Recruitment site where the sample was collected: Nanoro (Burkina Faso), Navrongo (Ghana), Nairobi (Kenya), Agincourt (South Africa), Dikgale (South Africa), Soweto (South Africa).

**night-light—Night-time luminosity**. The average values of light pixels were obtained using night-light geo-tiffs from NOAA's NGDC Earth Observation Group (EOG): https://ngdc.noaa.gov/eog/ and spatial boundaries are taken from the Global Administrative Boundaries Database: http://gadm.org/.

**edu—Education**. The highest level of education was self-reported according to four categories: (1) No formal education, (2) Primary, (3) Secondary, (4) Tertiary. Tertiary education includes qualifications such as certificates, diplomas, or degrees.

**occu—Occupation**. Self-declared employment according to five categories. (1) Self-employed. (2) Formal full-time employment by someone else. (3) Part-time employment by someone else. (4) Informal employment (dependent on the availability of work). (5) Unemployed.

**SES.Q—Socio-economic-status quintiles**. The socio-economic status of the individual was calculated using a list of household goods appropriate for each site (not all sites included all variables as some items were not relevant in specific settings): electricity, solar energy, power generator, alternative power source, television, radio, motor vehicle, bicycle, refrigerator, washing machine, sewing machine, telephone, mobile phone, microwave, DVD player, satellite TV or DSTV, computer or laptop, internet by computer, internet by mobile phone, electric iron, fan, electric or gas stove, kerosene stove, plate gas, electric plate, torch, gas lamp, kerosene lamp with a glass, toilet facilities, potable water, grinding mill, table, sofa set, wall clock, bed, mattress, blankets, cattle, other livestock, poultry, tractor, plough. In order to establish a meaningful comparison between individuals at different sites, the SES quintiles were calculated, and individuals assigned to a quintile. This approach was used to remove a potential cultural component related to socio-economic-status[2].

Anthropometry variables

**Height—meters**. Standing height measured in meters.

**Weight—kg**. Weight measured in kilograms.

**BMI—Body mass index—kg/m²**. Weight in kilograms divided by height in meters squared.

**WHR– Waist: Hip ratio—no units**. Calculated by dividing the individual waist circumference in centimeters by the hip circumference also in centimeters.

**VAT—Visceral adiposity tissue—cm**. Visceral (medial) fat measured using ultrasound.

**SCAT—Subcutaneous adiposity tissue—cm**. Subcutaneous (transverse) fat measured using ultrasound.

Lipid composition (in fasting serum)

**LDL—Low-density lipoprotein cholesterol—mmol/L**.

**HDL—High-density lipoprotein cholesterol—mmol/L**.

**TC—Total cholesterol—mmol/L**.

**TG—Total triglycerides—mmol/L**.

Circulatory

**Systolic BP—Systolic blood pressure—mmHg**. Three readings were taken during a single session, two minutes apart, the first was discarded and the second two readings were averaged.

**Diastolic BP—Diastolic blood pressure—mmHg**. Three readings taken during a single session, the first was discarded and the second two readings were averaged.

**Pulse—Heart rate—beats per minute**. Three readings taken during a single session, the first was discarded and the second two readings were averaged.

**Assessing site economic and urban development**. As a proxy for the urban and economic development of the different AWI-Gen sites we used night-light intensity (luminosity). Luminosity has been widely used as a proxy in countries where GDP data are either not available or of poor quality[31,32]. It was found to be highly correlated with GDP per capita and other measures of prosperity, like electricity provision[33], and can therefore be considered as a valid proxy[34,35]. More recently, using Demographic and Health Surveys (DHS) from 29 African countries, night-time light intensity has been found to correlate strongly with indicators of household wealth, education, and health. Also, the variation in night-time light explained a substantial share in the variation of these indicators[36].

Night light data are made available by the National Geophysical Data Center (NGDC) of the National Oceanic and Atmospheric Administration of the US, and originate from images taken by satellites of the Defense Meteorological Satellite Program (DMSP) of the U.S. Department of Defense between 1992 and 2013. Night light intensity data are available on pixel (grid cell) level, with each pixel corresponding to $30 \times 30$ arc s, i.e., one value represents the average night light intensity of an area of 0.86 square kilometers (on the equator). Night light intensity is measured by an integer ranging from 0 (unlit) to 63. We use the latest version (4.0) of the data.

**ROH calling**. ROH longer than 300 Kb were called using PLINK software with the following parameters:

*homozyg-snp 30*. Minimum number of SNPs that a ROH is required to have.
*homozyg-kb 300*. Length in Kb of the sliding window.
*homozyg-density 30*. Required minimum density to consider a ROH (1 SNP in 30 Kb).
*homozyg-window-snp 30*. Number of SNPs that the sliding window must have.
*homozyg-gap 1000*. Length in Kb between two SNPs in order to be considered in two different segments.
*homozyg-window-het 1*. Number of heterozygous SNPs allowed in a window.
*homozyg-window-missing 5*. Number of missing calls allowed in a window.
*homozyg-window-threshold 0.05*. Proportion of overlapping window that must be called homozygous to define a given SNP as in a "homozygous" segment.

No linkage disequilibrium pruning was performed. These conditions were previously used and validated by different published studies and were shown to call ROH that correspond to autozygous segments in which all SNPs (including those not present on the array) are homozygous-by-descent[9,28,37].

**Calculating genomic inbreeding coefficients**. The inbreeding coefficient or $F_{IT}$, is defined as the probability that an individual receives two alleles identical-by-descent[38]. Traditionally, $F_{IT}$ was measured using deep genealogies; currently, we can obtain an estimate of this parameter without having any genealogy by using genomic approaches. Different genomic inbreeding coefficients were calculated.

$F_{ROH}$ measures the actual proportion of the autosomal genome that is autozygous over and above a specific minimum length ROH threshold[39]. When analyzing ROH $> 1.5$ Mb, $F_{ROH}$ correlates most strongly ($r = 0.86$) with the inbreeding coefficient obtained from an accurate six-generation pedigree[39]. Using extended pedigrees of the royal European dynasties, with complex inbreeding loops, it has been found that above the 10th generation the change in the inbreeding coefficient is less than 1%[40]. Also, it has been found that individuals with no inbreeding loops in at least 5 generations (and probably 10) carried ROH up to 4 Mb in length but not longer[39]. $F_{ROH}$ using a genomic approach, captures the total inbreeding coefficient ($F_{IT}$) within the resolution of the data available and the size of ROH that can be called[9].

$$F_{ROH} = \frac{\sum_{i=1}^{n} ROH > 1.5Mb}{3\,Gb} \tag{3}$$

$F_{GRM}$. An alternative genomic inbreeding coefficient was obtained using PLINK's parameter -ibc (Fhat3). This coefficient described by Yang et al. 2011 ($\hat{F}^{III}$)[41] is defined as:

$$F_{GRM} = \frac{1}{N} \sum_{i}^{n} \frac{\left(x_i^2 - (1 + 2p_i)x_i + 2p_i^2\right)}{2p_i(1 - p_i)} \tag{4}$$

where N is the number of SNPs, $p_i$ is the reference allele frequency of the ith SNP and $x_i$ is the number of copies of the reference allele. The reference allele frequencies were site-specific and included only loci with MAF $> 0.05$.

Clark et al. 2020 showed that $\hat{\beta}_{F_{GRM}}$ is downwardly biased in real data and that this is proportional to the ratio $var(F_{ROH})/var(F_{GRM})$, as expected when the difference between $F_{GRM}$ and $F_{ROH}$ can be considered as estimation error. In the same work, they compared the relative abilities of $F_{ROH}$ and $F_{GRM}$ to capture inbreeding using the pedigree information of 47927 Icelanders with a 10 generation pedigree. The correlation was highest for $F_{ROH}$ ($r = 0.779$) in comparison to $F_{GRM}$ ($r = 0.682$).

$F_{outsideROH}$ An additional genomic inbreeding measure, $F_{outsideROH}$, was calculated as the genomic fraction of homozygous SNPs outside ROH.

$$F_{outsideROH} = \frac{O\prime(HOM) - E\prime(HOM)}{N\prime - E\prime(HOM)} \tag{5}$$

Where:

$$O\prime(HOM) = O(HOM) - N_{SNP\_ROH} \tag{6}$$

$$E\prime(HOM) = \left(\frac{N - N_{ROH}}{N}\right) * E(HOM) \tag{7}$$

$$N\prime = N - N_{ROH} \tag{8}$$

where $O(HOM)$ is the observed number of homozygous SNPs, $E(HOM)$ is the expected number of homozygous SNP according to H-W proportions, N is the

total number of non-missing genotyped SNPs and $N_{SNP\_ROH}$ is the number of homozygous SNPs found in ROH.

**Testing inbreeding depression**. Inbreeding depression (ID) was modeled by a multiple regression: $y = \beta_{FROH} * F_{ROH} + Xb + \varepsilon$ (9). Where $y$ is a vector of measured trait values, $\beta_{FROH}$ is the unknown scalar effect of $F_{ROH}$. $F_{ROH}$ is a known vector of individual $F_{ROH}$ as described above, $b$ is a vector of unknown covariates effects including the mean ($\mu$). $X$ is a known design matrix for the fixed effects, and $\varepsilon$ is an unknown vector of residuals. The traits used in this study have been the subject of genome-wide association meta-analysis (GWAMA), phenotype modeling, such that inclusion of covariates were chosen with reference to leading consortia: The Genetic Investigation of Anthropometric Traits (GIANT) for anthropometry, Global Lipids Genetics Consortium (GLGC) for lipids composition International Consortium for Blood Pressure (ICBP) for blood pressures.

SES.Q = sex + age + edu + occu + night_light + pc1 + … + pc15
Height = sex + age + age$^2$ + SES.Q + night_light + pc1 + … + pc15
Weight = sex + age + age$^2$ + SES.Q + night_light + pc1 + … + pc15
BMI = sex + age + age$^2$ + SES.Q + night_light + pc1 + … + pc15
WHR = sex + age + age$^2$ + height + SES.Q + night_light + pc1 + … + pc15
VAT = sex + age + age$^2$ + height + SES.Q + night_light + pc1 + … + pc15
SCAT = sex + age + age$^2$ + height + SES.Q + night_light + pc1 + … + pc15
HDL = sex + age + age$^2$ + SES.Q + night_light + pc1 + … + pc15
LDL = sex + age + age$^2$ + SES.Q + night_light + pc1 + … + pc15
TC = sex + age + age$^2$ + SES.Q + night_light + pc1 + … + pc15
TG = sex + age + age$^2$ + SES.Q + night_light + pc1 + … + pc15
Pulse = sex + age + age$^2$ + SES.Q + BMI + night_light + pc1 + … + pc15
Systolic BP = sex + age + age$^2$ + SES.Q + BMI + night_light + pc1 + … + pc15
Diastolic BP = sex + age + age$^2$ + SES.Q + BMI + night_light + pc1 + … + pc15

In order to account for unmeasured confounding variables that may differ between rural and urban sites, we used three different lines of evidence. First, socio-economic-status was added as a confounding variable to every model we ran. This allows the model to account for differences in the relative socio-economic status within sites and across individuals in all sites. Second, the luminosity was added to prevent further confounding effects due to differential urban development. Third, when the overall dataset was stratified by luminosity we added "site" as a co-variable and we tested the effect of $F_{ROH}$ on SES.Q for both luminosity groups. We expect that, if there are no confounding variables introducing bias, the effect of the $F_{ROH}$ over SES.Q should be similar in both groups.

To analyze inbreeding depression, and for computational efficiency, it was decided to solve the multiple regression models in two steps. In the first step, the trait ($y$) was regressed on all fixed covariates to obtain the maximum likelihood (ML) of the model: $y = Xb + u + \varepsilon'$ (10), where $u$ is an unknown vector of polygenic effects with multivariate normal distribution of mean 0 and covariance matrix $\sigma_g^2 A$, where A is the genomic relationship matrix (GRM). GRM was obtained using PLINK v1.9 and GRM[42], residuals ($\varepsilon'$) were estimated using GenABEL[43]. These residuals were used in subsequent analyses. To estimate $\beta_{FROH}$ for each trait, trait residuals were regressed on $F_{ROH}$ to obtain the ML solution of the model: $\varepsilon' = \mu + \beta_{FROH} * F_{ROH} + \beta_{F\_outsideROH} * F_{outsideROH} + \varepsilon$ (11). Sex-specific estimates of $\beta_{FROH}$ and $\beta_{F\_outsideROH}$ (shown in Tables S1–S4) were obtained from this model applied to the relevant sex. Also, specific estimates of $\beta_{FROH}$ and $\beta_{F\_outsideROH}$ were obtained for sites with more and less than 5 night-light units. When comparing between sites with more and less than 5 night-light units, night-light as a covariate was removed from the different models, but the site of sampling was added as a covariate in order to account for potential site confounding effects.

Finally, in order to be able to compare inbreeding effects among traits and sexes, $\beta_{FROH}$ were standardized using each trait's standard deviation.

**Assessing whether inbreeding depression is caused by common or rare variants**. $F_{ROH}$ is an estimate of autozygosity which increases the homozygosity of all variants, both common and rare. In contrast, $F_{GRM}$ is calculated from common SNPs (>5%MAF) and correlates well with the homozygosity of common SNPs but less with rare SNP which may be in weak LD. Following previous studies[10] and in order to assess if ID is caused by common or rare variants, we performed bivariate models of all traits (Trait ~ $F_{ROH}$ + $F_{GRM}$) to establish whether the observed inbreeding effects associate more strongly with $F_{ROH}$ or $F_{GRM}$.

By analyzing the excess homozygosities of SNPs, extracted from the UK Biobank, at seven minor allele frequencies, Clark et al.[10] showed that homozygosity of common SNPs is better predicted by $F_{GRM}$, but rare variant homozygosity is better predicted by $F_{ROH}$.

**Measuring genome-specific regional Inbreeding Depression**. In order to learn more about the genetic architecture of each trait, the effect of ROH burden was tested in each of ~1000 3 Mb-wide windows along the genome. For each window, the fraction which is in ROH was calculated, and then the association between ROH in each 3 Mb window and the mean trait residual was tested. A Bonferroni correction for 1000 windows was applied, significance was considered when $P < 5 \times$

$10^{-5}$. Results are shown in a Manhattan-plot in Fig. 4. QQ plots are shown in Supplementary Fig. 1.

**Reporting summary**. Further information on research design is available in the Nature Research Reporting Summary linked to this article.

## Data availability
AWI-Gen phenotype dataset is available at dataset accession id: EGA00001002482. AWI-Gen genotype dataset accession id: EGAD00010001996.
**Night-time luminosity**. The average values of light pixels were obtained using night-light geo-tiffs from NOAA's NGDC Earth Observation Group (EOG): https://ngdc.noaa.gov/eog/ and spatial boundaries are taken from the Global Administrative Boundaries Database: http://gadm.org/. Data used correspond to the 2013 version, last accessed on February 10, 2020.

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

## Acknowledgments

F.C.C. received a postdoctoral fellowship from the South African National Research Foundation. The AWI-Gen Collaborative Centre is funded by the National Human Genome Research Institute (NHGRI), Office of the Director (OD), Eunice Kennedy Shriver National Institute of Child Health & Human Development (NICHD), the National Institute of Environmental Health Sciences (NIEHS), the Office of AIDS research (OAR) and the National Institute of Diabetes and Digestive and Kidney Diseases (NIDDK), of the National Institutes of Health (NIH) under award number U54HG006938 and its supplements, as part of the H3Africa Consortium. Additional funding came from the Department of Science and Technology, South Africa, award number DST/CON 0056/2014. M.R. is a South African Research Chair in Genomics and Bioinformatics of African populations hosted by the University of the Witwatersrand, funded by the Department of Science and Technology, and administered by the National Research Foundation. J.F.W. acknowledges support from the UK MRC Human Genetics Unit quinquennial programme grant (MC_UU_00007/10).

## Author contributions

F.C.C.: research plan development, data analysis, figures, and manuscript preparation; S.H.: advisor on genomic analysis, data analysis, and manuscript preparation; D.C.: data analysis. GoAg, GeAs, P.B., X.G.O., F.M., and S.N.: AWI-Gen dataset collection and preparation; M.R.: principal advisor for research plan development, data analysis, and manuscript preparation; J.W.: advisor for research plan development, data analysis, and manuscript preparation. All authors read and approved the final manuscript.

## Competing interests

The authors declare no competing interests.
