## [Peer Review File · Nature Communications]

REVIEWER COMMENTS

Reviewer #1 (Remarks to the Author):

The authors use a large dataset of human genetic data and medical records from eight sites in Africa to study the effects of inbreeding on several measures associated sometimes with cardiometabolic diseases. The study is fairly well described and seems to be conducted well. However, there are various aspects that are insufficiently described to be properly assessed. It seems like the authors used good care to avoid pitfalls of multiple testing (although a few aspects should be made clearer in the manuscript to be sure). Overall, I think this will be an interesting study, and I have a fairly small number of specific comments below (with numbers referring to line numbers).

Title: I am not sure if it is helpful to include the abbreviation of a consortium in the title. I suggest to remove affiliations from the title.

29 and 38-39: Overdominance also leads to inbreeding depression, not only directional dominance.

32: What does AWI-Gen stand for? I suggest to avoid abbreviations without explanations, and am not sure if this is needed in the abstract.

64: Other than selection pressure, there are also other explanations for recessiveness of deleterious alleles. For example, one functional copy of a gene may often be sufficient to provide almost the full functionality and thus slow a process down just a little bit.

65: This sentence suggests that runs of homozygosity are the only genomic measures of inbreeding. However, although ROH may be the best measure, there are other well-used ones too.

74: The authors seem to suggest a causal relationship of inbreeding affecting socioeconomic status. What is this direction of causality based on? It seems conceivable that socioeconomic status affects mating behaviors and mate availability and thus a reverse causality.

86: What is meant with "TG levels, which are not significant..."? It seems like the analysis would show if FROH was a significant predictor of TG levels, and not the other way around as suggested in this phrasing.

91: How exactly was this tested? Was an interaction between light intensity and FROH tested here, as would seem necessary? Please explain in more detail.

91-92: Why would a significant result be the consequence of lack of power? Lack of power would be expected to explain the failure to find a significant result.

96: confounding by what? Please explain/expand.

100-101: What were these quintiles used for? Where are the results for the quintile analysis? As is now, I find this sentence confusing, and I do not know what to take away from it. Are the results above somehow based on analysis of quintiles?

106: Is this across all traits or only the four traits mentioned on line 103?

121-122: Please explain why this is the case. Surely, ROHs will be homozygous for both common and rare alleles, so how does this result imply that (i) rare alleles are responsible, and (ii) that directional dominance rather than overdominance is responsible?

138: What is the evidence for directional dominance over overdominance?

335: Quite a bit more information is needed here, and a brief explanation/justification for the equations would be helpful too. For example, what is N and NROH?

338: According to the verbal description, the equation should start with $bFROH * FROH$ (not $bFROH + FROH$).

343: What is GWAMA? How were those covariates selected by the consortia? As it is, it is not replicable, so please explain in detail unless a publicly available reference can be given. For example, why is age sometimes included as quadratic variable and sometimes not, or why is height included as a covariate for some but not all models?

344: What are these principal components of ancestry? Please explain and add references if needed.

346-359: What does the "-" stand for in the formulas? Should that be "="?

371-372: Please explain how beta was standardized. Division by SD or anything else?

457: Is this a regular Bonferroni correction, i.e. using $0.05/14$ for each trait? Or is it some sort of sequential Bonferroni correction? I suggest to clarify.

Supporting information:

Please explain how socioeconomic status was estimated. The current description would not allow to replicate this, or apply in a similar situation.

Reviewer #2 (Remarks to the Author):

1. While the findings of the present study in SSA populations is novel and interesting, ROH studies on cardiometabolic traits have not been successful in other ancestral populations even with large sample sizes. There are major issues that need to be addressed to ensure robustness of findings (given absence of replication cohorts), also considering the expectation that ROH are relatively fewer in SSAs compared to these other populations where the effect of IDs on many quantitative traits was found to be null.
2. Due to the nature of the tests for detecting ROH across genomic segments, genetic structure may bias ROH estimations and could complicate stability of models that define ROH window boundaries, and consequently their associations with traits. I am concerned that the ROH and ID in this study may have been confounded by pooling of ancestrally diverse SSA populations from wide biogeographic backgrounds into a single analytic sample. The authors need to characterize the population structure and demographic history of the samples (e.g. plots of PCs by region/site etc) and provide justification in support of pooling these samples together. Moreover, the extent to which the burden of ROH is influenced by population structure needs to be examined; correction by adjusting for PCs should only be considered a secondary step.
3. There are several layers of differences between the two SES strata defined based on luminosity. These include differences in geographic distribution, sample size, and demographic histories, and population genetic structure (west-southern vs east-Southern). The contributions of these variations to the observed differences in the effect of ID on traits needs further investigation.
4. On page 4, the authors mentioned 'selective pressure' from lifestyle factors may explain their observations. Given urbanization (and the associated lifestyle dynamics) is a recent phenomenon in SSA, it does not appear to qualify as a selective force that could change the allele frequency structure of the genome.
5. Was the study adequately powered to conclude "For all other traits we can conclude that no-major loci impart the effect, rather the ID is polygenic in origin"?

Reviewer #3 (Remarks to the Author):

The study examines the effect of distal inbreeding on cardio-metabolic traits within a large and diverse African cohort. They see a number of traits affected by Froh, and show that these effects are mediated by sex as well as urban/rural development. These results further establish the effect of inbreeding on cardio-metabolic traits by investigating a population that had not been strongly represented in previous studies of distal inbreeding.

While the general results and analysis are well done and follow from previously published studies, I think there needs to be a number of additional figures and analyses to confirm the nature and veracity of the results. Below are my main points:

- 1) Use the actually samples size after QC (10,617) instead of rounding to 10k
- 2) With 1.7m SNPs, and calling ROH > 300 kb with no LD pruning, I'm worried there are ROH segments that are much older floating around. How did you check this? Were there any frequency checks? Could this explain the weaker signal in more urban areas?
- 3) I suggest adding a "long ROH" analysis. Either longer segments (> 1Mb) and/or using LD pruning.

Alternatively, examining the significant traits after removing Froh outliers can tell us how much they are driving the results. Importantly, how does sex / luminosity fare as mediators when looking at longer ROH or removing outliers?

4) More direct comparison to the Clark et al (2019) results are needed. What is consistent? what isn't? I think the hypothesis is that directional dominance should be consistent across populations, and if not, what could be the explanation? A sign test of direction for the 13 traits would be good.

5) A forest plot by collection site for each significant trait would be instrumental in knowing how each African site is contributing to the signal. Are the results being driven by a single site with higher mean Froh?

6) The violin plots in Fig 1 are pretty, but I think leave out a lot of key information. Does mean Froh differ by site? Does the variance differ by site?

7) Similar to 6, how does the urban/rural split look in terms of mean Froh / variance?

8) Follow up more on the sex difference signal - is this seen in other studies/populations? How could recombination explain this? Is there a mean difference in Froh by sex? Difference in variance? Is one site driving this result? Important to exclude as many explanations/confounders as you can with the data in hand if this result is inconsistent with previous results.

9) For ROH mapping, following up on the significant signals is needed. A recessive SNP test in the region could narrow which gene is driving the result. Or a ROH breakpoint test, where each start/end is tested for association.

10) Presuming your alpha is 0.05, the bonferroni correction for each trait should be $5e-5$, not $1e-5$. However, with multiple traits tested, this can change. Just be clear about it

11) "It was interesting to consider whether the genome-wide ID effects" should simply be "We tested for genome-wide ID effects"

12) How were PCs calculated? Within site? After sites were combined? How do PCs associate to Froh?

We thank all three reviewers for their insightful comments and suggestions. We have added text, improved explanations and corrected errors, all of which have improved the manuscript.

Reviewer #1

Title: I am not sure if it is helpful to include the abbreviation of a consortium in the title. I suggest to remove affiliations from the title.

We would prefer to retain the name of the consortium in the title, but if the editor would like it removed we would be happy to do that.

29 and 38-39: Overdominance also leads to inbreeding depression, not only directional dominance.

Thank you pointing this out, we have changed the text to add overdominance as a possible explanation for inbreeding depression.

32: What does AWI-Gen stand for? I suggest to avoid abbreviations without explanations, and am not sure if this is needed in the abstract.

This study is part of a larger cohort study, but we agree that it can be confusing to have the abbreviation in the abstract. We have changed the text and added the full name: Africa Wits-INDEPTH partnership for Genomic studies.

64: Other than selection pressure, there are also other explanations for recessiveness of deleterious alleles. For example, one functional copy of a gene may often be sufficient to provide almost the full functionality and thus slow a process down just a little bit.

We agree with the reviewer that there may be other possible explanations for the recessiveness of deleterious alleles. However, we are not describing the events that make a deleterious allele recessive, but the factors that can influence the effect size of the inbreeding depression. These are mainly two: the amount of inbreeding in the population and the selection pressure, as was shown by Falconer & Mackay (1996), Crnokrak & Roff (199), Bataillon & Kirkpatrick (2000) and Kirkpatrick & Jarne (2000).

65: This sentence suggests that runs of homozygosity are the only genomic measures of inbreeding. However, although ROH may be the best measure, there are other well-used ones too.

We have changed the sentence. We added “For this study” to emphasize that using ROH was our own choice.

74: The authors seem to suggest a causal relationship of inbreeding affecting socioeconomic status. What is this direction of causality based on? It seems conceivable that socioeconomic status affects mating behaviors and mate availability and thus a reverse causality.

Thank you for pointing out the possibility for misunderstanding. We have separated out the socio-economic relationship into another sentence. We did not mean to infer a causal relationship; there are indeed many different scenarios where a relationship can arise between socio-economic status and mating behaviours.

It now reads as follows: The substantial levels of genomic homozygosity exhibited by the AWI-Gen cohort allow us to explore their effect on 13 CMD-associated traits and their relationship to socio-economic status (Figure 2a). We detected a negative association of inbreeding depression with socio-economic-status within each site; in general, having a higher F_{ROH} was associated with a lower socio-economic-status.

86: What is meant with “TG levels, which are not significant...”? It seems like the analysis would show if F_{ROH} was a significant predictor of TG levels, and not the other way around as suggested in this phrasing.

We thank the reviewer for noticing this mistake. We have changed the manuscript in order to reflect that the non-significant test is the inbreeding depression.

It now reads: Moreover, TG levels, for which significant ID was not detected after multiple test correction in the overall analysis, are strongly significantly affected by ID in the study sites with less night-light (0.024 mmol/L).

91: How exactly was this tested? Was an interaction between light intensity and F_{ROH} tested here, as would seem necessary? Please explain in more detail.

Thank you for pointing out the need to explain this more carefully. This was tested by comparing the inbreeding depression slopes between the urban (sites with night light above than 5 units) and rural (sites with night light intensity below 5 units) sites using an ANCOVA – Analysis of Covariance. These slopes are shown in Figure 2b. These analyses were done using R. Night Light intensity was incorporated as a confounding variable in the overall analysis, so its effect on the traits was accounted for. By doing this we prevent night light intensity from confounding the inbreeding depression analysis (Figure 2a) without the need to estimate an interaction between F_{ROH} and night light intensity. Moreover, we did not detect correlation between night light intensity and the F_{ROH} among the different sites: R²= 0.02 pval=0.0987. Finally we split sites according to their night light intensity (above and below 5 night-light units), when doing this we removed light night intensity as a covariable from the model but we added “site” to address for differences between sites among the same class of night light intensity. In order to get the test provided at line 91 we compare regression slopes between sited with more and less than 5 night-light units.

We have added text to the methods to make this more explicit, as follows: Assessing Site Economical and Urban Development: As a proxy for urban and economic development of the different AWI-Gen sites we used night-light intensity (luminosity). Luminosity has been widely used as a proxy in countries where GDP data are either not available or of poor quality^{30,31}. It is found to be highly correlated with GDP per capita and other measures of prosperity, like electricity provision³², and can therefore be considered as a valid proxy^{33,34}.

91-92: Why would a significant result be the consequence of lack of power? Lack of power would be expected to explain the failure to find a significant result.

We thank the reviewer for noticing this mistake. We have deleted this sentence.

96: confounding by what? Please explain/expand.

Thank you for pointing out this omission. We have altered the text to clarify that we mean unmeasured confounding variables, that differ between rural and urban sites. Since we find that there is a negative association between the inbreeding coefficient and the social-economic-status in both rural and urban sites, this suggests that our analysis is not being affected by uncontrolled confounding variables. We would expect a different outcome if some confounding variable were introducing some bias between rural and urban sites. We provide three different lines of evidence that our results are not due to unmeasured confounding variables.

- 1. Socio-economic status was added as a confounding variable to every model we ran. This allows the model to account for differences in socio-economic status between sites and individuals**
- 2. Night light intensity was also added to prevent from further confounding effects of urban development**
- 3. When the overall dataset was stratified by Night Light intensity, the effect of the F_{ROH} over the socio-economic questionnaire variable was in the same direction and similar effect for both strata.**

100-101: What were these quintiles used for? Where are the results for the quintile analysis? As is now, I find this sentence confusing, and I do not know what to take away from it. Are the results above somehow based on analysis of quintiles?

We have clarified this in the text and refer the reader to the methods section and supplementary notes where it is explained how the quintiles were derived. Social-economic-status of each individual was calculated using a catalogue of household goods and deriving quintiles within each site (it was not done across all sites, as the relevant household goods were country-specific). In order to compare this across regions and cultures of our sampling sites, we therefore used the within-site quintiles. This is a routine process when comparing populations around Africa with great heterogeneity in household goods. Further explanation can be found in the supplementary Note that provides the trait definitions.

The following was added to the methods: Besides the above biomarkers a socio-economic-status variable was also obtained for each individual. This variable (SES) was obtained by adding the items a person has in their household from a defined number of items, determined as appropriate for their country. The SES quintiles for each study site were calculated and individuals assigned to a quintile. This was used in the analysis to remove a potential cultural component related to socio-economic-status. See Supplementary Notes for trait definitions.

106: Is this across all traits or only the four traits mentioned on line 103?

This is correct, it is for all traits, indeed the Mann-Whitney test was obtained by comparing all the effects between men and women. We have clarified this by adding “across all traits”

It now reads: The mean effect across all traits of a homozygous genome (β_{FROH}) was 4.9 ± 5.9 Trait_{SD} in men and only 1.1 ± 3.8 Trait_{SD} in women (Mann-Whitney U test $p = 0.0155$).

121-122: Please explain why this is the case. Surely, ROHs will be homozygous for both common and rare alleles, so how does this result imply that (i) rare alleles are responsible, and (ii) that directional dominance rather than overdominance is responsible?

We want to thank the reviewer for raising this important issue. We have changed the manuscript to address it and added a new section in the online material and methods to explain how we concluded that rare alleles are responsible for the inbreeding depression.

Basically, in order to assess whether common or rare variants are responsible for the inbreeding depression we compared the effects of the F_{ROH} and F_{GRM} . F_{GRM} is a measurement of the genomic inbreeding coefficient that considers only common variants (MAF > 5%). By fitting a bivariate model we show that the conditioned effect of $\beta_{\text{FROH}} | F_{\text{GRM}}$ is of a greater magnitude than $\beta_{\text{FGRM}} | F_{\text{ROH}}$, and further, that for most of the traits $\beta_{\text{FGRM}} | F_{\text{ROH}}$ does not differ from 0.

The following was added: Assessing whether Inbreeding Depression is caused by common or rare variants. F_{ROH} is an estimate of autozygosity which increases the homozygosity of all variants, both common and rare. In contrast F_{GRM} is calculated from common SNPs (>5%MAF) and contrast well with the homozygosity of common SNPs but less with rare SNP which may be in weak LD. Following previous studies¹⁰ and in order to assess if ID is caused by common or rare variants, we performed bivariate models of all traits ($\text{Trait} \sim F_{\text{ROH}} + F_{\text{GRM}}$) to establish whether the observed inbreeding effects associate more strongly with F_{ROH} or F_{GRM} .

By analysing the excess homozygosities of SNPs, extracted from the UK Biobank, at seven minor allele frequencies, Clark et al. 2019 showed that homozygosity of common SNP is better predicted by F_{GRM} , but rare variant homozygosity is better predicted by F_{ROH} .

138: What is the evidence for directional dominance over overdominance?

We believe that our data support directional dominance, for the following reasons: Overdominance – that is positive selection on heterozygotes bringing alleles to intermediate frequency – would predict that more common homozygous SNPs outside long ROH would also have an effect. The fact that the bivariate models show that F_{ROH} captures the signal better than F_{GRM} and that $F_{\text{outsideROH}}$ shows little effect, are both evidence that it is directional dominance on rarer alleles that is likely to explain the inbreeding depression we observe here.

335: Quite a bit more information is needed here, and a brief explanation/justification for the equations would be helpful too. For example, what is N and NROH?

Thank you for pointing out the need to be more explicit in the equations to ensure they are well understood by the reader. We have added additional explanations in the manuscript for clarity.

338: According to the verbal description, the equation should start with $b_{FROH} * FROH$ (not $b_{FROH} + FROH$)

Thank you for pointing this out, the manuscript has been corrected.

343: What is GWAMA? How were those covariates selected by the consortia? As it is, it is not replicable, so please explain in detail unless a publicly available reference can be given. For example, why is age sometimes included as quadratic variable and sometimes not, or why is height included as a covariate for some but not all models?

Thank you for pointing us to these omissions. We updated the manuscript and included more information regarding the covariates selection, which followed the best practices of the leading consortia focused on these traits. We have also fixed some minor errors in the covariates included in each model.

344: What are these principal components of ancestry? Please explain and add references if needed.

These are standard principal components of genetic variation introduced by Price (2006) and used in genetic association studies since 2007 (Wellcome Trust Case Control Consortium 2007) to control for population structure.

346-359: What does the “-“ stand for in the formulas? Should that be “=”?

Thank you, we changed the manuscript accordingly.

371-372: Please explain how beta was standardized. Division by SD or anything else?

This is now explained as follows: “Finally, in order to be able to compare inbreeding effects among traits and sexes, β_{FROH} were standardized using each trait’s standard deviation.” We divided the slope (each trait’s regression beta) by the standard deviation of the same trait.

457: Is this a regular Bonferroni correction, i.e. using 0.05/14 for each trait? Or is it some sort of sequential Bonferroni correction? I suggest to clarify.

Yes, this is a regular Bonferroni: 0.05/14, and is thus conservative. We changed the manuscript accordingly. The following was added to the legend of Figure 2: “A regular Bonferroni correction for multiple testing was applied; significance is shown with an asterisk (0.05/14).”

Supporting information:

Please explain how socioeconomic status was estimated. The current description would not allow to replicate this, or apply in a similar situation.

For clarity, we point the reader to the Supplementary material: trait definition.

Reviewer #2

1. While the findings of the present study in SSA populations is novel and interesting, ROH studies on cardiometabolic traits have not been successful in other ancestral populations even with large sample sizes. There are major issues that need to be addressed to ensure robustness of findings (given absence of replication cohorts), also considering the expectation that ROH are relatively fewer in SSAs compared to these other populations where the effect of IDs on many quantitative traits was found to be null.

We understand the reviewer's concern about robustness and replication; however we wish to call attention to a larger study published by Clark et al. (Nat Comms 2019). They analysed over 1.4 Million individuals for a large number of traits including cardiometabolic risk factors such as BMI, plasma lipids, blood pressure, etc and showed association with ROH which we replicate and extend here. While Clark et al were able to include 20,000 individuals of sub-Saharan African descent, all but 1000 individuals are from the American Continent and thus we present a statistically-powered ROH analysis of Africans for the first time. Our study and Clark et al.'s paper share in total 12 traits (height, weight, BMI, hips to waist ratio, total cholesterol, HDL, LDL, fat percentage, triglycerides, heart rate, diastolic blood pressure and systolic blood pressure). For all these traits the direction of the effect was the same for both studies. For 5 out of 12 traits the FROH effects (the inbreeding depression) were significant in both studies. Furthermore, both our and Clark et al.'s studies found that men suffer stronger effects from inbreeding depression in comparison to women, and that for several traits the inbreeding depression of men and women are of opposite sign. We have replicated these results.

Finally, we have shown that SSA has ROH burden, especially for ROH>1.5Mb (Ceballos et al. 2019. Human Genetics). In the current manuscript, we show there is sufficient variance to detect ID. In common with all studies, the fact that we observe no effect does not mean there is not a small effect present, but the point here is to highlight the traits for which reasonably large effects are seen.

2. Due to the nature of the tests for detecting ROH across genomic segments, genetic structure may bias ROH estimations and could complicate stability of models that define ROH window boundaries, and consequently their associations with traits. I am concerned that the ROH and ID in this study may have been confounded by pooling of ancestrally diverse SSA populations from wide biogeographic backgrounds into a single analytic sample. The authors need to characterize the population structure and demographic history of the samples (e.g. plots of PCs by region/site etc) and provide justification in support of pooling these samples together. Moreover, the extent to which the burden of ROH is influenced by population structure needs to be examined; correction by adjusting for PCs should only be considered a secondary step.

We appreciate these comments and spent much time thinking about this and taking care to analyse the data in a way that would justify pooling the data.

- The ROH calling and the calculation of the genomic inbreeding coefficient FROH have been shown to not be affected by population structure. ROH calling through PLINK follows an observational approach where variant calling quality and SNP density are the important factors to consider (see Ceballos et al. Nature Reviews Genetics 2019).
- Population structure may be important for the analysis of the inbreeding depression when dealing with different populations. However, by introducing in the model 15 principal components of ancestry as covariables, the model incorporates possible effects of ancestry. The estimations of the slopes we provide already have incorporated the possible effect of population structure, as is standard practice in quantitative human genetics.
- Adding principal components of ancestry to the model is a routine practice in genome wide analysis like GWAS or ID analysis. Introduced by Price (2006) and used in genetic association studies since 2007 (Wellcome Trust Case Control Consortium 2007) to control for population structure.
- Specifically in sub-Saharan Africa, MalariaGEN (an international community of researchers working to understand how genetic variation in humans, Plasmodium parasites and anopheles mosquitos affects the biology and epidemiology of malaria) have been integrating in genomic studies data from different parts of Africa by adding PCs to control for population structure. In their last paper published last year they integrated genome-wide data of 17,000 individuals from Africa, Asia and Oceania by accounting for population structure in their association model using the first 10 PCs. (Malaria Genomic Epidemiology Network, Nat Comm. 2019).

3. There are several layers of differences between the two SES strata defined based on luminosity. These include differences in geographic distribution, sample size, and demographic histories, and population genetic structure (west-southern vs east-southern). The contributions of these variations to the observed differences in the effect of ID on traits needs further investigation.

We are aware of the heterogeneity of the socio-economic status of the different sites. However, we wish to indicate that we took this into account by:

- We have used night-light intensity in different ways. We first added it as a covariable in the general model in order to account for urban development differences between different sites. In the overall model we also add as covariables the socio-economic-status and the first 15 PCs. We believe, and it has been shown by previous studies, that this is enough to account for differences in geographic distribution, demographic histories and population genetic structure among sub-Saharan Africans.
- We then split the AWI-Gen cohort into two classes according to the night-light intensity (above and below 5 light-night units). In order to account for geographical distribution we added the “site” as a covariate. In this sub-analysis we also added the socio-economic-status and 15 PCs in order to account for cultural and ancestry differences between individuals.
- Our motivation to add these covariables is not to study the relationship between the traits and the night-light intensity or the different PCA, but to allow our multiple

regression models to remove these effects from the analysis of the inbreeding depression. This is standard procedure when dealing with linear models.

4. On page 4, the authors mentioned 'selective pressure' from lifestyle factors may explain their observations. Given urbanization (and the associated lifestyle dynamics) is a recent phenomenon in SSA, it does not appear to qualify as a selective force that could change the allele frequency structure of the genome.

We agree with the reviewer that this issue needs further consideration. Urbanisation has taken place over a period of several generations and affects the phenotype - more than the alleles frequencies and genome structure. We believe that this may be causing some gene-environment effects. We have modified our manuscript in order to express that the different environment (as we are not sure if this is mediated by a difference in selection pressure) is behind the differential inbreeding depression between urban and rural sites.

We have changed the sentence as follows: We suggest that environmental changes affecting lifestyle factors may be playing a role since food resources, comfort commodities and even primary healthcare are limiting factors in less developed sites.

5. Was the study adequately powered to conclude "For all other traits we can conclude that no-major loci impart the effect, rather the ID is polygenic in origin"?

Thank you for your relevant comment. Indeed we cannot rule out lack of power to detect significant ID within the genomic windows analysed. We have changed the text as follow: "For all other traits we have no evidence of major loci that may be exerting large effects, rather the ID appears to be polygenic in origin."

Reviewer #3

1) Use the actual samples size after QC (10,617) instead of rounding to 10k

We have mentioned the maximum actual sample size (N=10,776 before and 10,617 after QC), however not all participants have complete data for each variable and therefore the actual number of individuals included differs according to trait, as is shown in the supplementary tables.

2) With 1.7m SNPs, and calling ROH > 300 kb with no LD pruning, I'm worried there are ROH segments that are much older floating around. How did you check this? Were there any frequency checks? Could this explain the weaker signal in more urban areas?

Thank you for highlighting this concern. We have looked at ROH longer than 300kb but we did not use this threshold for the Inbreeding depression analysis. We first estimated Inbreeding depression using ROH longer than 1.5Mb where LD structures are of little importance (F_{ROH}) and we compared it with the inbreeding depression obtained using ROH shorter than 1.5Mb ($F_{outsideROH}$) which will include LD structures. Our results show that it is the ROH longer than 1.5b that is provoking the inbreeding depression in the corresponding traits. These results were also constant across urban and rural areas.

There were no significant effect of $F_{\text{outsideROH}}$ for any trait.

3) I suggest adding a "long ROH" analysis. Either longer segments (> 1Mb) and/or using LD pruning. Alternatively, examining the significant traits after removing Froh outliers can tell us how much they are driving the results. Importantly, how does sex / luminosity fare as mediators when looking at longer ROH or removing outliers?

Thank you for raising the potential concern. We addressed this by analysing ROH longer than 1.5 Mb. Dealing with regression analysis the variance of the axes are extremely important for the power of the analysis. Indeed with more extreme values of FROH is possible to detect stronger effects over the phenotypes. This is because of the leverage of the regression model. We do not have FROH outliers in the statistical sense: we analyse this using the Inter Quartile range and Tukey's fences $[Q1-k(Q3-Q1), Q3+k(Q3-Q1)]$, no individual was larger than $k=1.5$, showing that there were no outliers in our dataset. Our analysis mediated by sex or by urbanization showed that ROH longer than 1.5Mb are the ones responsible for the inbreeding depression. In any case slopes obtained using the $F_{\text{outsideROH}}$ as an explanatory variable were not significant. This is telling us that long ROH (ROH>1.5Mb) are the ones responsible for the inbreeding depression. This outcome replicates that of Clark et al. 2019. However, the relative low burden of ROH over 5Mb in Sub-Saharan Africa prevents us from doing further analysis.

4) More direct comparison to the Clark et al (2019) results are needed. What is consistent? what isn't? I think the hypothesis is that directional dominance should be consistent across populations, and if not, what could be the explanation? A sign test of direction for the 13 traits would be good.

Thank you for this suggestion of a more detailed comparison and we agree that it is expected that directional dominance would be consistent.

We changed the text on pages 3 and to provide more comparisons with Clark et al. 2019. Directional dominance and even sex-specific effects are consistent across populations.

Our study and Clark et al.'s paper share in total 12 traits (height, weight, BMI, waist to hip ratio, total cholesterol, HDL, LDL, fat percentage, triglycerides, heart rate, diastolic blood pressure and systolic blood pressure). For all these traits the effect was directionally consistent between both studies. For 5 out of 12 traits (height, weight, total cholesterol, BMI and triglycerides) the effect of the F_{ROH} (the inbreeding depression) was significant in both studies. Furthermore, both our and Clark et al. 's studies found that men suffer stronger inbreeding depression in comparison to women, and even that for several traits the inbreeding depression of men and women are of opposite sign. We have replicated these results.

The following was added: "As a replication comparison, we can conclude that the effect of the inbreeding depression has the same direction for the 12 traits shared by Clark et al 2019¹⁰ and this study. Furthermore, among the African-descendant individuals analysed by Clark et al. 2019¹⁰, 5 out of 12 traits F_{ROH} were found to be significant by both studies."

5) A forest plot by collection site for each significant trait would be instrumental in knowing how each African site is contributing to the signal. Are the results being driven by a single site with higher mean Froh?

This is a good question and we have tested differences of the F_{ROH} between sites. We have found that they are not that different as can be seen in figure 1. While there is insufficient power at any one site, inspection of the point estimates of beta reveals that the signal is not being driven by only one site. This can be seen in the following table:

	SES.Q		WEIGHT		BMI		SCAT		TG	
	Beta	P-val	Beta	P-val	Beta	P-val	Beta	P-val	Beta	P-val
Agincourt	-12.67	0.016	-115.08	0.078	-38.06	0.10	-7.97	0.331	-1.42	0.48
Dikgale	-10.58	0.067	-98.5	0.101	-40.3	0.09	-8.35	0.412	-1.52	0.32
Nairobi	-7.58	0.051	-32.2	0.21	-12.3	0.21	1.23	0.52	1.35	0.52
Soweto	-5.21	0.125	-12.3	0.56	-9.32	0.42	3.24	0.641	0.35	0.65
Nanoro	-10.25	0.042	-135.21	0.052	-25.6	0.251	-9.21	0.085	-2.12	0.087
Navrongo	-11.12	0.035	-87.1	0.065	-21.7	0.19	-10.21	0.074	-1.85	0.11

6) The violin plots in Fig 1 are pretty, but I think leave out a lot of key information. Does mean Froh differ by site? Does the variance differ by site?

This is an important question and we used the violin plots to show the distribution of the F_{ROH} . Although we are aware that it does not capture all the information about ROHs. Since the F_{ROH} across sites has a strong positive skew, obtaining its mean and variance for the different sites is not very informative. The differences in the median are not very large: Nanoro = 0.0026; Dikgale = 0.0062; Soweto = 0.0061; Navrongo = 0.0034; Agincourt = 0.0072; Nairobi = 0.0042. If we classify by sex we also find small differences: men = 0.0045, women = 0.0049. Finally if we classify by night-light intensity: urban sites (night light > 5 units) = 0.0051, rural sites (night light < 5 units) = 0.0047.

We have further added a sentence to the paper to clarify that the urban and rural sites do not differ in F_{ROH} (medians 0.0051 vs 0.0047) as follows: “No differences in the F_{ROH} were found between men and women as also between urban and rural sites (F_{ROH} men = 0.0045, women = 0.0049) (F_{ROH} urban sites = 0.0051, rural sites = 0.0049).”

We also added the underlying medians and interquartile ranges to supplementary table 7

7) Similar to 6, how does the urban/rural split look in terms of mean Froh / variance?

As now detailed, the medians are very similar: urban sites (night light > 5 units) = 0.0051, rural sites (night light < 5 units) = 0.0047

8) Follow up more on the sex difference signal - is this seen in other studies/populations? How could recombination explain this? Is there a mean difference in Froh by sex? Difference in variance? Is one site driving this result? Important to exclude as many explanations/confounders as you can with the data in hand if this result is inconsistent with previous results.

Thank you for highlighting this important issue. We observed no strong difference between the median F_{ROH} for men and women.

Our results replicate those in Clark et al 2019. The FROH effects over the 12 traits shared by ours and Clark et al.'s paper are directionally consistent. As mentioned above, for 5 out of 12 traits (height, weight, total cholesterol, BMI and triglycerides) the effect of the F_{ROH} (the inbreeding depression) was significant in both studies. Finally, the same differences between the inbreeding depression in males and females were found by both studies. We believe recombination is not the explanation, but the difference in genetic architecture and gene-environment interaction that influences the phenotype. This hypothesis requires further study that will be tackled in the future.

9) For ROH mapping, following up on the significant signals is needed. A recessive SNP test in the region could narrow which gene is driving the result. Or a ROH breakpoint test, where each start/end is tested for association.

This would indeed be interesting and important. We are, however, not doing ROH mapping but just analysing the inbreeding depression using the homozygosity in 100 genomic windows regardless if these windows include or not a whole ROH. We agree that it will be very interesting to pursue ROH mapping in larger more powerful datasets in the future.

10) Presuming your alpha is 0.05, the bonferroni correction for each trait should be $5e-5$, not $1e-5$. However, with multiple traits tested, this can change. Just be clear about it

Thank you, this has been corrected in the manuscript.

11) "It was interesting to consider whether the genome-wide ID effects" should simply be "We tested for genome-wide ID effects"

Thank you, we have changed this sentence.

12) How were PCs calculated? Within site? After sites were combined? How do PCs associate to Froh?

PCs were generated using PLINK from the entire data set after LD-pruning. We generate PCs from the entire data set since we are adjusting for global population structure. Our intention is not to study the relationship between FROH and PC. We add 15 PC to the co-variables when dealing with the first part of the model (see paper's material and methods) residuals obtained from this model were regressed against the FROH. Our purpose by adding 15 PCs is to incorporate in the model any effect of population structure.

REVIEWERS' COMMENTS:

Reviewer #1 (Remarks to the Author):

The authors have made changes and updates to the manuscript, but I still think the manuscript needs clarifications and better descriptions at various places. It also seems that the recent additions and modifications have somewhat impacted the flow, readability and clarity of the manuscript. So I think a more thorough overhaul of the text would be nice to make sure all sections are consistent with what is included in this current version.

Specific comments (lines refer to the version with tracked changes in red):

32-33: I still think it would be more informative for most reader if the 10 words used to promote the consortium (which is not informative to a general audience) were for example used to say where the study sites are specifically.

85-87: Check commas and traits (e.g. "last three" on line 84 implies that more than three traits should be listed above; delete "experiencing")

87-92: Please reference a table with the results of this study, and ideally also those of Clark, so that the reader can directly compare effect sizes and also see what traits are referred to (rather than just saying e.g. 5 (unnamed) traits had the same effect).

94-96: A statistical test would be needed to make such a statement, but is still lacking. As suggested before, testing for an interaction of light intensity and FROH would be a straightforward way to test for that for a specific trait. Simply comparing point estimates is not sufficient (and even then, the pattern across all traits is not at all as clear as this sentence suggests).

96-97: The statement "average reduction in trait value was greater for each of the traits" (assuming "greater in dark sites" is missing here) is not consistent with Figure 2b (see e.g. SES.Q, Pulse, etc.).

97: reductions for what amount of increase in FROH?

102-104: Where is the description for this ANCOVA analysis? Based on the response to reviews letter, it seems clear that it would not include a test of an interaction. This leads me to believe that these P-values likely refer to the effect of light intensity on the trait (e.g. trait difference in dark and light sites), and not to the effect of light intensity on inbreeding depression (as this paragraph is about and the writing would suggest also here). Please add more information, and clarify by specifying exactly what effect size is referenced here (line 102). A table with ANCOVA results somewhere is required too.

136: The introduction emphasized that only FROH (and not FGRM) was used in this study, so it comes as a surprise to read about FGRM now.

312: The reference to Supplementary Notes here is not helpful, because these notes don't contain any additional information about socioeconomic status that is not also described here. Rather, lists of these household items for each country and references to publications that describe these lists should be added (e.g. line 309).

385: What is used for reference allele frequencies? The frequencies across the whole dataset or the frequencies in a given population? This FGRM is sensitive to allele frequencies because it is designed to give higher weight to rarer homozygotes. As a consequence, the method likely doesn't work very well with differentiated populations.

431-432: So far, SES.Q was described as having been calculated WITHIN sites (i.e. with quantiles calculated by considering only the values for a specific site). If that is still true, it can NOT be used to correct for socioeconomic differences among sites as claimed here.

435: What means "tested effect of FROH over SES.Q for both cohorts"? Please explain and/or rephrase.

445: This equation is different from the one on line 402, and not suitable to estimate effects of FROH as they are currently presented (i.e. without emphasizing the caveat that FoutsideROH is also in the models). The whole approach in this section is now fairly confusing to follow, because the paragraphs following lines 402 initially suggest that a quite different approach was taken.

456: This restriction to common SNPs should already be mentioned when FGRM is described earlier.

455-458: This sentence needs a reference.

Figure 2:

655: Do you mean 95% confidence intervals? Or what are "interval coefficients"?

Figure 2b: Are the bars in the plot illustrating standard errors or confidence intervals?

Figure 3: Please specify what the bars are illustrating.

Reviewer #2 (Remarks to the Author):

The authors have addressed my comments. I encourage them to present the PCA plots annotated by site/urban-rural status at least in the supplementary file.

Reviewer #3 (Remarks to the Author):

I am generally satisfied with the responses to my review of the manuscript.

We thank you reviewer for the helpful comments and have addressed them below.

32-33: I still think it would be more informative for most reader if the 10 words used to promote the consortium (which is not informative to a general audience) were for example used to say where the study sites are specifically.

As the reviewer perhaps has noticed we already show the sites at figure 1A (New line 54)

We would prefer to retain our 10 words explaining and promoting the consortium and we believe this is informative for the audience. In part to counterbalance the Eurocentric bias in genomic studies to date, it is important to promote the first genomic consortium being developed in sub-Saharan Africa. We already placed study sites in a map in figure 1. This figure is referenced in the text.

85-87: Check commas and traits (e.g. “last three” on line 84 implies that more than three traits should be listed above; delete “experiencing”

This section now reads as follows: “When the entire cohort is considered, we detected significant negative ID for weight, BMI and SCAT. However, the intensity of the ID for these traits was not very strong and the offspring of the equivalent of a second cousin mating had on average, a reduction of 0.51 kg, 0.18 kg/m², and 0.049 cm, respectively.” (New lines number 89 – 91).

87-92: Please reference a table with the results of this study, and ideally also those of Clark, so that the reader can directly compare effect sizes and also see what traits are referred to (rather than just saying e.g. 5 (unnamed) traits had the same effect).

We obtained these estimates by using the regression slopes of Supplementary Table 1 and 2. Clark’s regression slopes estimates can be found in his Supplementary Data 18. We have made specific reference to this in the text.

We also changed the text, included the references for the tables, and we specify the traits we are referring too. (New lines 95 – 98).

94-96: A statistical test would be needed to make such a statement, but is still lacking. As suggested before, testing for an interaction of light intensity and FROH would be a straightforward way to test for that for a specific trait. Simply comparing point estimates is not sufficient (and even then, the pattern across all traits is not at all as clear as this sentence suggests).

We appreciate the reviewer’s comment, however we believe that there is enough evidence to support the statement that the inbreeding depression is more intense in sites with lower night light intensity (NL), for the following reasons. First of all, it is important to notice that, unlike when sites with NL<5 are considered, we did not detect any significant inbreeding depression for any trait analysed in sites with NL>5 (Figure 2B). Slopes (effect sizes: BetaFROH) in traits like weight, BMI, SCAT, LDL and TG are significant in the NL<5 sub-cohort but not in the NL>5 sub-cohort. This is important considering that the sample size and the FROH distribution are very similar between those sites. In addition, although we are dividing the dataset into two groups which entails a reduction in statistical power, effects sizes and p-

values are more extreme in the NL<5 sub-cohort in comparison to the analysis of the complete dataset. In our view the pattern is clear, since we can only assess this for the traits that exhibit statistically significant inbreeding depression (weight, BMI, SCAT, LDL and TG), for the rest of the traits the slopes can't be considered different from 0.

We agree with the reviewer that a valid method to test if a categorical variable may produce differences in slopes is to test the interaction of that categorical variable with the continuous variable. However, this is not the only approach. It is also possible to split the database by the categorical variable and compare both regression models through an ANOVA. Both approaches are acceptable.

96-97: The statement "average reduction in trait value was greater for each of the traits" (assuming "greater in dark sites" is missing here) is not consistent with Figure 2b (see e.g. SES.Q, Pulse, etc.).

We agree that this sentence needs clarity. We only considered the traits for which the effect sizes (slopes) were statistically significantly different from 0 (Weight, BMI, SCAT).

To clarify, we modified the sentence as follows: "The average reduction in trait value for offspring of the equivalent of second cousin mating was greater for each of the significantly associated traits in areas with less luminosity (0.70 kg in weight, 0.26 kg/m² in BMI and 0.081 cm in SCAT)."

97: reductions for what amount of increase in FROH?

See sentence above, now referring to second cousin mating.

102-104: Where is the description for this ANCOVA analysis? Based on the response to reviews letter, it seems clear that it would not include a test of an interaction. This leads me to believe that these P-values likely refer to the effect of light intensity on the trait (e.g. trait difference in dark and light sites), and not to the effect of light intensity on inbreeding depression (as this paragraph is about and the writing would suggest also here). Please add more information, and clarify by specifying exactly what effect size is referenced here (line 102). A table with ANCOVA results somewhere is required too.

We did not include an explanation of the ANCOVA procedure since it is a routine process to compare whether two regression slopes are different. We compared regression models using R (aov() R command). In the text we cite the method and show the correspondent p-values.

We tested the interaction between the FROH and the NL for the different traits (table below). The interaction between the FROH and NL was significant for SCAT, LDL and TG but not for SES.Q. To perform this analysis, we first regressed the traits on all fixed covariates except the NL. We then regressed these residuals on FROH + NL and on FROH:NL (interaction). ANOVA comparisons between models with and without interactions were also significant except for the SES.Q trait. This shows that slopes are significantly different for these traits between regions with low and high NL intensity.

Trait	F value	P-value
SES.Q		
FROH	13.54	1.5E-03
NL	0.01	0.909
FROH:NL	0.125	0.724
SCAT		
FROH	20.65	5.17e-06
NL	343.2	<2e-16
FROH:NL	15.28	9.3e-05
LDL		
FROH	6.43	0.0112
NL	343.22	<2e-16
FROH:NL	4.769	0.0290
TG		
FROH	11.88	0.005
NL	251.51	<2e-16
FROH:NL	17.95	2.2e-05

136: The introduction emphasized that only FROH (and not FGRM) was used in this study, so it comes as a surprise to read about FGRM now.

We thank the reviewer for raising this important issue. We emphasized FROH since it has been used successfully in inbreeding depression analyses in different published studies (Clark et al. 2019, Yoshi et al. 2015, McQuillan et al. 2012) and is the gold standard in ROH analysis (Ceballos et al. 2018).

This analysis, with the FGRM, was done along with the rest of the analysis, in first place, but not included in the first submitted manuscript. The reviewer made a solid point and we realize that we need to add them. However, the inclusion of this new inbreeding coefficient, in the second version of the manuscript, does not change our outcomes. Is just to show that rare alleles are responsible for the inbreeding depression.

312: The reference to Supplementary Notes here is not helpful, because these notes don't contain any additional information about socioeconomic status that is not also described here. Rather, lists of these household items for each country and references to publications that describe these lists should be added (e.g. line 309).

Following the reviewer and editor's recommendation, we have added to the methods section the trait definition. We also included the list of the household items that were considered to obtain the SES variable. (New lines 277 – 287).

385: What is used for reference allele frequencies? The frequencies across the whole dataset or the frequencies in a given population? This FGRM is sensitive to allele frequencies because it is designed to give higher weight to rarer homozygotes. As a consequence, the method likely doesn't work very well with differentiated populations.

Thank you for requesting clarification. The allele frequencies were obtained for each site and this is clarified in the methods section. Since FGRM is obtained using high frequency alleles

(MAF > 0.05) we expect that the differences in allele frequencies would not bias in the calculation of FRGM.

431-432: So far, SES.Q was described as having been calculated WITHIN sites (i.e. with quantiles calculated by considering only the values for a specific site). If that is still true, it can NOT be used to correct for socioeconomic differences among sites as claimed here.that why

We do not compare *between* sites and do not have the data that allows us to do so. Rather we compare *between individuals* using their relative SES status as the metric. Using quintiles based an asset register of household goods is common practice in comparing *individual* wealth in heterogeneous data, as we already cited in the main text. Using quintiles allow a *relative* comparison between individuals affected by heterogeneity when comparing different populations. In this case we use the quintiles of the socio-economic-status to assess if inbreeding has any effect on the socio-economic-status of a given inbred individual in comparison to outbred individuals within own populations. This trait is therefore comparable among individuals in heterogenous populations since we are not directly comparing the SES but the relative socio-economic-status of an individual relative to his or her population.

435: What means “tested effect of FROH over SES.Q for both cohorts”? Please explain and/or rephrase.

We rephrased the sentence as follows: Third, when the overall dataset was stratified by luminosity we added “site” as a co-variable and we tested the effect of F_{ROH} on SES.Q for both luminosity groups.

445: This equation is different from the one on line 402, and not suitable to estimate effects of FROH as they are currently presented (i.e. without emphasizing the caveat that FoutsideROH is also in the models). The whole approach in this section is now fairly confusing to follow, because the paragraphs following lines 402 initially suggest that a quite different approach was taken.

We thank the reviewer for this comment. It is correct the equations are not related.

Equation 5 describes the estimation of the genomic inbreeding coefficient outside ROH (FoutsideROH). To analyse inbreeding depression, and for computational efficiency, we used a multiple regression analysis solved in two steps. We first regressed the trait against all fixed covariates as per the equation 10. We regressed the residuals obtained in this first step in a second step where we regressed them on FROH + FoutsideROH, in equation 11.

455-458: This sentence needs a reference.

We have included the reference. (New line 439).

Figure 2:
655: Do you mean 95% confidence intervals? Or what are “interval coefficients”?
Figure 2b: Are the bars in the plot illustrating standard errors or confidence intervals?
Figure 3: Please specify what the bars are illustrating.

We thank the reviewer for noticing this mistake. As is indicated in the label on the X-axis the bars illustrate the 95% confidence intervals. We have changed the text accordingly. We changed “95% interval coefficient” to “95% confidence intervals”. (New line 606)